# Novel 1,8-Naphthalimide Derivatives Inhibit Growth and Induce Apoptosis in Human Glioblastoma

**DOI:** 10.3390/ijms252111593

**Published:** 2024-10-29

**Authors:** Cheng-Chi Lee, Chuan-Hsin Chang, Yin-Cheng Huang, Tzenge-Lien Shih

**Affiliations:** 1Department of Neurosurgery, Chang Gung Memorial Hospital, Linkou Medical Center, Chang Gung University, Taoyuan 333423, Taiwan; yumex86@gmail.com; 2Department of Research, Taipei Tzu Chi Hospital, Buddhist Tzu Chi Medical Foundation, New Taipei City 231016, Taiwan; chuanhsin032484@gmail.com; 3Research Center for Chinese Herbal Medicine, Graduate Institute of Healthy Industry Technology, College of Human Ecology, Chang Gung University of Science and Technology, Taoyuan 333324, Taiwan; 4Department of Chemistry, Tamkang University, Tamsui Dist., New Taipei City 251301, Taiwan

**Keywords:** 1,8-Naphthalimide derivatives, human glioblastoma, anticancer

## Abstract

Given the rapid advancement of functional 1,8-Naphthalimide derivatives in anticancer research, we synthesized these two novel naphthalimide derivatives with diverse substituents and investigated the effect on glioblastoma multiforme (GBM) cells. Cytotoxicity, apoptosis, cell cycle, topoisomerase II and Western blotting assays were evaluated for these compounds against GBM in vitro. A human GBM xenograft mouse model established by subcutaneously injecting U87-MG cells and the treatment responses were assessed. Both compounds **3** and **4** exhibited significant antiproliferative activities, inducing apoptosis and cell death. Only compound **3** notably induced G2/M phase cell cycle arrest in the U87-MG GBM cells. Both compounds inhibited DNA topoisomerase II activity, resulting in DNA damage. The in vivo antiproliferative potential of compound **3** was further validated in a U87-MG GBM xenograft mouse model, without any discernible loss of body weight or kidney toxicity noted. This study presents novel findings demonstrating that 1,8-Naphthalimide derivatives exhibited significant GBM cell suppression in vitro and in vivo without causing adverse effects on body weight or kidney function. Further experiments, including investigations into mechanisms and pathways, as well as preclinical studies on the pharmacokinetics and pharmacodynamics, may be instrumental to the development of a new anti-GBM compound.

## 1. Introduction

1,8-Naphthalimides can be synthesized from 1,8-naphthanhydrides through a reaction with corresponding amines, as depicted in Figure 1 and the Appendix A. Research has indicated that 1,8-Naphthalimide derivatives exhibit various biological activities, including antitumor [1], anti-inflammatory, and antiviral properties [2,3]. Novel synthetic 1,8-Naphthalimide derivatives have been developed and have been demonstrated to possess biological activities such as antiplatelet and antithrombotic effects. Therefore, these derivatives may be candidates for the development of novel antiplatelet agents for the treatment of cardiovascular diseases [4]. In addition to having anticancer properties, 1,8-Naphthalimide derivatives have been discovered to have applications in various material science fields. Consequently, they have been employed in chemical sensors [5] for the detection of toxic waste [6,7,8], metal ions [9], and anions [9]. Moreover, 1,8-Naphthalimide-based compounds have been employed as real-time cellular imaging agents, enabling real-time monitoring of cellular processes [10,11,12]. Furthermore, 1,8-Naphthalimide-based compounds have demonstrated aggregation-induced emission or aggregation-induced emission enhancement effects that lead to the production of bright colors in their solid states [13,14,15,16,17]. This unique phenomenon renders them valuable for applications in optical devices [18]. Moreover, these compounds exhibit anticancer properties, particularly against murine melanoma [4,19].

The 1,8-Naphthalimide chromophore is a planar structure that intercalates with DNA base pairs, leading to DNA denaturation and subsequent apoptosis of tumor cells [2]. Because DNA topoisomerase II (Top II) plays a key role in DNA cleavage, drugs targeting DNA intercalation can effectively inhibit Top II activity [20]. Moreover, the overexpression of Top II in various tumor types renders it a promising molecular target for the development of anticancer agents [21]. Previous studies have reported that novel 1,8-Naphthalimide derivatives and C4-benzazole Naphthalimide derivatives exhibit antimelanoma activity through the inhibition of human DNA Top II activity [19,22]. As shown in Figure 1, representative examples of anticancer 1,8-Naphthalimide derivatives include mitonafide [23] and amonafide [24]. Mitonafide is synthesized by reacting 1,8-naphthanhydride with *N*,*N*-dimethylethylenediamine, followed by nitration [4,25]. Amonafide is derived from the reduction of mitonafide by SnCl_2_ in HCl [25]. Both of these drugs have been applied in clinical trials [26,27]. Additionally, novel 1,8-Naphthalimide derivatives have been demonstrated to induce DNA damage and autophagic cell death in lung cancer cells through the inhibition of the human demethylase fat mass and obesity-associated protein (FTO), resulting in suppressed multicellular spheroid growth of lung cancer cells [28]. In a previous study, we reported the synthesis of compounds **1** and **2** for evaluating murine melanoma [19]. Compound **1** demonstrated promising activity as a DNA Top II inhibitor [19], leading to reduced platelet activation and thrombus formation [4]. As shown in Figure 2 and in the Appendix A, we further synthesized compounds **3** and **4** using similar methods. Our previous work outlines the synthetic procedure [19], with detailed spectroscopic data provided in the Appendix A. The C4 linker of the 1,8-Naphthalimide moiety was modified to incorporate ethylenediamine rather than the piperidine moiety because such an alteration could produce a more potent inhibitor of DNA Top II [5]. The ongoing progress in designing anticancer drugs based on 1,8-Naphthalimides has been extensively reviewed in the literature [2,29,30,31]. However, studies using 1,8-Naphthalimide derivatives on glioblastoma multiforme (GBM) are lacking. The aim of our study is to investigate the in vitro and in vivo effects of 1,8-Naphthalimide derivatives on GBM.

## 2. Results

### 2.1. Cytotoxic Effects

The anticancer activities of compounds **3** and **4** against GBM cell viability were evaluated. A 48 h assessment of their direct anticancer effects was conducted using two GBM cell lines: U87-MG and DBTRG-05MG. As shown in Table 1, the IC_50_ values of compounds **3** and **4** for the U87-MG cells were 11.11 ± 1.63 and 30.48 ± 1.86 μM, respectively. Additionally, compounds **3** and **4** exhibited potent cytotoxic activities against DBTRG-05MG, with IC_50_ values of 5.58 ± 1.30 and 17.32 ± 1.40 μM, respectively.

### 2.2. Clonogenic Ability of Compounds **3** and **4**

A colony formation assay was employed to investigate the effects of compounds **3** and **4** on the proliferation of the U87-MG GBM cells, with TMZ serving as the positive control. Figure 3 illustrates that the clonogenic ability of cells treated with either compound **3** or **4** was significantly diminished in U87-MG at a dose of 5 μM (*p* < 0.001). Additionally, an in vitro cell survival assay of colony formation was performed to assess the effects of compounds **3** and **4** on another GBM cell line, DBTRG-05MG. As depicted in Appendix A, the clonogenic ability of the DBTRG-05MG GBM cells treated with either compound **3** or **4** was significantly reduced at doses of 5, 10, and 25 μM (*p* < 0.001 for all).

### 2.3. Induction of Apoptosis and Cell Death in GBM Cells by Compounds **3** and **4**

Apoptosis induction was detected using TACS Annexin V–FITC apoptosis detection kits (Bd Pharmingen; Bd Biosciences, Milpitas, CA, USA). The cellular apoptosis and death in the U87-MG GBM cells were categorized into three groups on the basis of the intensity of Annexin V and propidium iodide (PI) fluorescence: early apoptosis (Annexin V–FITC^+^/PI^−^ quadrant), late apoptosis (Annexin V–FITC^+^/PI^+^ quadrant), and cell death (Annexin V–FITC^+^/PI^+^ quadrant; Figure 4). Treatment of the human U87 GBM cells with 5 μM of compound **3**, 5 μM of compound **4**, or 50 μM of TMZ (as a positive control) significantly induced apoptosis in the U87-MG cells (33.00% ± 21.70%, 48.40% ± 20.31%, and 56.50% ± 23.80%, respectively; Figure 5). Additionally, the treatments with compounds **3** and **4** significantly promoted cell death in the U87-MG cells (55.00% ± 16.70% and 15.18% ± 11.46%, respectively; Figure 4).

### 2.4. Effects of Compounds **3** and **4** on the Inhibition of GBM Cell Proliferation Through G2/M Arrest

To investigate the effects of compounds **3** and **4** on cell cycle phases, the U87-MG GBM cells were treated with either compound **3** or compound **4**, and the cell cycle progression was examined. The cells were exposed to 5 μM of either compound **3** or compound **4** for 24 h, and the results revealed that only treatment with compound **3** increased the number of cells in the G2/M phase for the U87-MG GBM cells (Figure 5A,C). TMZ (50 μM) was used as a positive control. The DBTRG-05MG cells were treated with either compound **3** or compound **4**, and the cell cycle phase distribution was analyzed. Consistent with the findings for the U87-MG cells, only compound **3** significantly induced G2/M phase arrest in the DBTRG-05MG cells (Appendix A) and thereby inhibited tumor cell proliferation.

### 2.5. Inhibitory Effects of Compounds **3** and **4** on Top II and Ability to Induce DNA Damage

As depicted in Figure 6, the inhibitory activities of compounds **3** and **4** were evaluated using a topoisomerase II drug screening kit, with etoposide (VP-16) serving as the positive control. The results indicated that Top II was inhibited in compound **3**, as evidenced by the appearance of linear DNA at a high dose (100 μM) and supercoiled DNA at low doses (1 and 10 μM). Compound **4** demonstrated a similar effect at both high and low concentrations.

To investigate whether the inhibitory effects of compounds **3** and **4** on Top II activity are associated more with DNA DSBs or damage, as indicated by the phosphorylation of H2AX, the U87-MG GBM cells were treated with compound **3**, compound **4**, or TMZ, which served as a positive control. The results revealed that treatment with compounds **3** and **4** (at doses of 5 μM) as well as 100 μM TMZ significantly induced H2AX phosphorylation in the U87-MG cells (Figure 7). Additionally, DBTRG-05MG GBM cells treated with compound **3** (at doses of both 5 and 10 μM), a high dose of compound **4** (at 5 and 10 μM), and TMZ (at doses of 100 and 500 μM) exhibited marked H2AX phosphorylation (γH2AX; Appendix A).

### 2.6. In Vivo Evaluation of the Anticancer Effects of Compounds **3** and **4** on U87-MG Mouse Xenograft Models

To further evaluate the in vivo antitumor activities of compounds **3** and **4**, a human U87-MG GBM mouse xenograft model was employed. Once the tumors reached a measurable size, each group was administered the vehicle control, compound **3** (5 mg/kg), or compound **4** (5 mg/kg) three times per week for 21 days, with the TMZ treatment serving as the positive control. The results revealed that treatment with compounds **3** and **4** significantly suppressed tumor growth compared with the vehicle control (*p* < 0.001; Figure 8A,C and Appendix A), and the in vivo antitumor efficacy of compound **3** was comparable to that of TMZ. Additionally, compound **3** was determined to have a greater potential tumor suppression ability. Notably, neither compound **3** nor compound **4** treatment had any discernible effect on body weight (Figure 8B). Additionally, neither compound **3** nor compound **4** treatment altered the serum levels of blood urea nitrogen (BUN) or creatinine (CRE) in the mice (Figure 8D,E), suggesting neither the individual nor the combined treatments induced kidney toxicity.

## 3. Discussion

GBM is a highly aggressive and invasive brain tumor associated with poor patient survival rates. The current standard of care for GBM involves surgical resection followed by a combination of radiation therapy and chemotherapy. Despite advancements having been made in these treatment modalities, the overall survival outcomes for patients with GBM remain dismal. The median survival time for GBM ranges from 12 to 16 months [32], with less than 20% of patients surviving beyond 1 year and less than 3% surviving beyond 3 years postdiagnosis [33]. Adopting aggressive therapeutic approaches is infrequently successful in treating the disease.

Adamantane was discovered in 1933 [34], and its applications in medicine were first described in 1973 [35,36,37]. This diamond-like hydrocarbon moiety has been incorporated into active agents to enhance their lipophilicity and pharmacological properties [38]. The adamantane group is often referred to as a lipophilic bullet and is widely used in drug design [39]. In compounds **2** and **3**, different groups are connected to the nitrogen atom within the imide moiety. Conversely, compound **4** has an arene group that replaces the adamantane moiety found in compound **3**. Studies have reported the use of 1,8-Naphthalimides for evaluating brain cancer. Compounds **2**–**4** are currently being investigated for their potential biological activities against glioblastoma cell lines.

As shown in Figure 3 and Appendix A, these findings corroborate the earlier results obtained from the CCK8 proliferation assay (Table 1), which indicated that the proliferation of cells treated with compound **3** or **4** was notably lower than that of U87-MG and DBTRG-05MG cells treated for 72 h. Accompanied by the clonogenic ability test, these results indicate that compounds **3** and **4** exhibited promising antiproliferative effects on the GBM cells. Apoptosis induction tests suggested that compounds **3** and **4** both induced apoptosis and cell death in the GBM cells. Moreover, cell cycle regulation plays a pivotal role in cancer cell proliferation and is associated with apoptosis [40]. Our results indicated that exposure to compound **3** increased the proportion of cells in the G2/M phase, and compounds **3** and **4** both reduced the proportion of cells in the G0/G1 phase in the GBM cells. This suggests that compound **3** may therefore have induced apoptosis through G2/M cell cycle arrest in the GBM cell lines. Obviously, compound **3** is a more potential anti-GBM agent in inducing cancer cell apoptosis and death.

Mammalian cells contain two types of topoisomerases: type I topoisomerases (TOP1, TOP1mt, TOP3α, and TOP3β), which induce single-strand breaks in DNA (SSBs), and type II topoisomerases (TOP2α, TOP2β, and SPO11), which induce double-strand breaks (DSBs) in DNA [41,42,43]. DSBs represent a severe form of DNA damage that can lead to the phosphorylation of H2AX at Ser139 (γH2AX), which accumulates at the site of damage within cells [44,45]. Additionally, the induction of DSBs during apoptotic cell death results in the formation of γH2AX [35]. Our research team previously reported that the primary anticancer mechanism of three C4-benzazole naphthalimide derivatives involves the inhibition of Top II activity [19,22]. We also demonstrated that C4-benzoxazole 1,8-phthalimide derivatives exhibit antiproliferative activities against murine melanoma through the inhibition of human DNA TOP II activity [19]. Notably, elevated expression levels of DNA TOP II alpha in GBM [36] have been associated with the Ki67 index [37], a marker of cell proliferation, and are significantly associated with tumor growth and poor survival outcomes in patients with GBM [37]. High expression of Topo-IIα has been proved to relate to a highly proliferative status in GBM. Temozolomide (TMZ) is the current first-line chemotherapy drug for GBM; it has been reported to provide a survival advantage among patients with glioblastoma [46,47]. TMZ has been identified as an inhibitor of human DNA TOP II alpha [37]. However, its effectiveness is limited in patients with nonmethylated MGMT (O6-methylguanine-DNA methyltransferase); moreover, TMZ is associated with some drug-related side effects [48]. 1,8-Naphthalimide derivatives may be a therapeutic alternative with potential due to their similar or superior efficacy and fewer side effects compared with TMZ. The current results, as shown in Figure 6, suggest that compounds **3** and **4** may both target Top II, which may contribute to their cytotoxic effects against cancer cells. As shown in Figure 7 and Appendix A, these data suggest that both compounds **3** and **4** induce DNA damage in GBM cells. Most importantly, compounds **3** and **4** significantly suppressed tumor growth in vivo, and the in vivo antitumor efficacy of compound **3** was comparable to that of TMZ. Notably, although both derivatives demonstrated tumor growth suppression, compound **3** exhibited superior efficacy compared with compound **4** in the animal study, with no effect on body weight and no kidney toxicity. These findings underscore the potential of these novel 1,8-Naphthalimide derivatives as anticancer agents because of their substantial cytotoxicity. In addition, compounds **3** and **4** share the same structure in their upper portions. The only structural difference between them is that compound **3** contains an adamantane group, while compound **4** contains a phenyl group. One of the main challenges in drug design in terms of the blood–brain barrier (BBB) is membrane penetration. Both memantine [49] and amantadine [50], which are used for the treatment of central nervous system (CNS) disorders, possess adamantane groups, suggesting potential future clinical applications in the treatment of CNS tumors.

This is the first study evaluating the in vitro and in vivo treatment responses when using 1,8-Naphthalimide derivatives on GBM cancer cells. The above findings strongly suggest that compounds **3** and **4** induce DNA damage by targeting Top II activity, thereby significantly inhibiting growth and inducing apoptosis in GBM cells. As a result, 1,8-Naphthalimide derivatives could act as innovative anticancer agents, and their combination with TMZ may offer a promising treatment strategy for GBM patients.

## 4. Materials and Methods

### 4.1. Chemicals and Reagents

Dulbecco’s Modified Eagle Medium (DMEM) and a 1× antibiotic–antimycotic solution were purchased from Promeg (Madison, WI, USA). Antihuman gH2AX, H2AX, cleaved PARP, cleaved caspase-3, Bax, Bcl-2, and GADPH antibodies were purchased from Cell Signaling Technology (Beverly, MA, USA). Horseradish peroxidase (HRP) antirabbit and mouse IgG was purchased from Thermo Fisher Scientific (Waltham, MA, USA). The Cell Counting Kit-8 (CCK8; 96992), myoinositol, folic acid, and the other chemicals were purchased from Sigma-Aldrich (St. Louis, MO, USA).

### 4.2. Cell Culture

The human glioblastoma cell lines U87-MG and DBTRG-05MG were obtained from the American Type Culture Collection (Manassas, VA, USA). Both GBM cell lines were cultured in DMEM (Corning, NY, USA) supplemented with 10% heat-inactivated fetal bovine serum (FBS) (Corning, NY, USA). The cells were incubated at 37 °C with 5% CO_2_ and were passaged twice weekly.

### 4.3. The Cytotoxicity Assay

Cell viability was assessed using the CCK8 assay. Briefly, 2 × 10^3^ to 2 × 10^4^ cells were plated into 96-well culture plates. Following a 24 h incubation period, the cells were treated with varying concentrations of the compounds (3.125, 6.25, 12.5, 25, 50, and 100 μM). After 72 h, the treated medium was centrifuged at 200× *g* for 5 min. TMZ (Sigma-Aldrich) served as a positive control against the GBM cells. The supernatant was incubated with CCK8 reagent (Sigma-Aldrich) for 1–4 h at 37 °C, and the absorbance was measured at 450 nm.

### 4.4. The Apoptosis Assay

To assess the apoptotic effects of compound **3** and **4** on the U87-MG and DBTRG-05MG GBM cells, the cells were labeled using the flow cytometry TACS Annexin V–FITC/propidium iodide–PE (PI-PE) apoptosis detection kit (R&D System, Minneapolis, MN, USA) in accordance with the manufacturer’s instructions. Briefly, the U87 and DBTRG-05MG cells in complete DMEM were seeded into 6-well plates at a density of at 5 × 10^5^ cells/well and incubated at 37 °C in the presence of 5% CO_2_ for 24 h. The cells were then treated with compound **3** and **4** or TMZ for 24 h. After, the cells were detached from the plates through trypsinization (500 μL of trypsin/well). Subsequently, 100 μL of TACS Annexin V and PI reagent were added to a 1.5 mL tube. The percentage of apoptotic and dead cells was determined using a flow cytometer (BD AccuriTM C6 Plus; BD Biosciences, Milpitas, CA, USA). The apoptosis experiment was performed in triplicate.

### 4.5. Colony Formation

U87-MG or DBTRG-05MG GBM cells were initially seeded at a density of 2 × 10^3^ cells per well into either 6- or 12-well plates. Subsequently, they were exposed to various concentrations of compound **3** (1, 5, 10, or 25 μM), compound **4** (1, 5, 10, or 25 μM), or TMZ (25 or 50 μM; as a positive control) in a medium without 10% FBS for 24 h. Following the 24 h treatment, the culture medium was replaced with DMEM containing 10% FBS. The culture medium was refreshed every 2 days without the addition of further compounds. After a 14-day incubation period, the visible colonies were washed with phosphate-buffered saline (PBS), fixed, and stained with 1% formalin containing 1% crystal violet. Colony formation was examined using an inverted microscope (Olympus Corporation, Tokyo, Japan).

### 4.6. Cell Cycle Assay

The cell cycle distributions of both human U87-MG and DBTRG-05MG GBM cells were analyzed using flow cytometry and PI staining. The cells were seeded at a density of 1 × 10^6^ cells/well into 6-well plates and treated with either 5 μM of compound **3** or 5 μM of compound **4** for 24 h. The treated cell groups were then fixed in cold 70% ethanol for 2 h. RNase A (60 μg/mL) and PI (50 μg/mL) were added into PBS, and the samples were incubated for 30 min in the dark at room temperature. The cell cycle distribution was then analyzed using flow cytometry (FACScalibur; BD Biosciences, Milpitas, CA, USA; Becton, Dickinson, and Company, Franklin Lakes, NJ, USA). A total of 10,000 cells were collected from each cell group, and the percentages of the cell populations in the sub-G0/G1, G2/M, and S phases were determined using Modfit LT version 2.0 software (Verity Software House, Topsham, Maine). The entire experiment was replicated thrice.

### 4.7. Top II Assay

The inhibitory effects of the compounds on Top II activity were evaluated using the topoisomerase II drug screening kit (TopoGen, Buena Vista, CO, USA) by following the manufacturer’s instructions. Reaction mixtures containing supercoiled pHOT-1 plasmid DNA, the Top II enzyme, and various concentrations of the indicated compounds were incubated at 37 °C for 30 min. A sample containing supercoiled DNA and Top II in the presence of 100 μM of etoposide VP-16 served as the positive control. Reactions were terminated through the addition of 10% sodium dodecyl sulfate, with this followed by proteinase K treatment for 15 min at 37 °C. Subsequently, the samples were subjected to electrophoresis on 1% agarose gel and stained with ethidium bromide. The DNA bands were then visualized under ultraviolet light. The experimental conditions were the same as those previously reported [22].

### 4.8. Western Blotting

A Western blot analysis was performed following a previously described method [51]. Briefly, the U87-MG or DBTRG-05MG (6 × 10^5^ cells) GBM cells were seeded into 6-well plates and cultured until 85–90% confluence was reached. Subsequently, the cells were treated with different concentrations (1, 5, and 10 μM) of compounds **3** and **4**. Thereafter, the cells were collected and lysed using radioimmunoprecipitation assay buffer. Lysates of total protein were separated through 12.5% sodium dodecyl sulfate–polyacrylamide gel electrophoresis and transferred onto polyvinylidene difluoride membranes. Blocking was performed, and the membranes were incubated with anti-H2AX, anti-γ-H2AX, and anti-GADPH primary antibodies (Cell Signaling Inc., Danvers, MA, USA) overnight at 4 °C. Subsequently, each membrane was incubated and shaken with horseradish peroxidase (HRP)-conjugated secondary antibodies at room temperature for 1 h. Finally, each membrane was visualized using an enhanced chemiluminescence (ECL) detection kit and the ImageQuant LAS 4000 Mini biomolecular imager (GE Healthcare, Woburn, MA, USA). The band densities were quantified using ImageJ software, version 1.53 (NIH, Bethesda, MD, USA).

### 4.9. Animal Study

Female *nonobese diabetic (NOD)/severe combined immunodeficient (SCID)* mice (BioLasco, Taiwan) aged 6–8 weeks old and weighing 29.19 ± 1.11 g were housed in plastic cages with ad libitum access to water and food. The animal experiments conducted in this study were approved by the Institutional Animal Care and Use Committee of Chang Gung Memorial Hospital (IACUC approval no.: 2023061303) and adhered to the recommendations of the Guide for the Use of Laboratory Animals [52]. To establish a human GBM xenograft mouse model, the mice were subcutaneously injected with 2 × 10^6^ U87-MG GBM cells into the right-side dorsa. When the tumors reached a volume of 150–200 mm^3^, the mice were randomly divided into groups, namely control, compound **3**, compound **4**, and TMZ groups. Compound **3** (5 mg/kg) or compound **4** (5 mg/kg) was injected intraperitoneally (i.p.) three times per week, and TMZ (12.5 mg/kg) was administered three times per week through oral gavage for 21 days. The tumor sizes were measured weekly, and the mice were euthanized when the tumors reached the maximum allowed volume. Additionally, their serum levels of BUN and CRE were determined using an automated clinical chemistry analyzer (Dri-Chem NX500i, Fujifilm, Tokyo, Japan).

### 4.10. Statistical Analysis

All the experiments were conducted in triplicate, and the data are presented as means ± standard errors of the mean. Statistical analysis was performed using Student’s *t* test (Prism, GraphPad Software 9.0.2, San Diego, CA, USA). In addition, *p* < 0.05 was considered significant.

## 5. Conclusions

A comparative analysis of the structures of compounds **3** and **4** revealed that the *N*,*N*-dimethylethylenediamine and adamantane moieties play crucial roles in these compounds’ biological activity. This study presents novel findings demonstrating that both derivatives exhibited significant GBM cell suppression in vitro and in vivo. Further experiments, including investigations into the mechanisms and pathways, as well as preclinical studies on the pharmacokinetics and pharmacodynamics, may be instrumental to the development of a new anti-GBM compound.

## Figures and Tables

**Figure 1 ijms-25-11593-f001:**
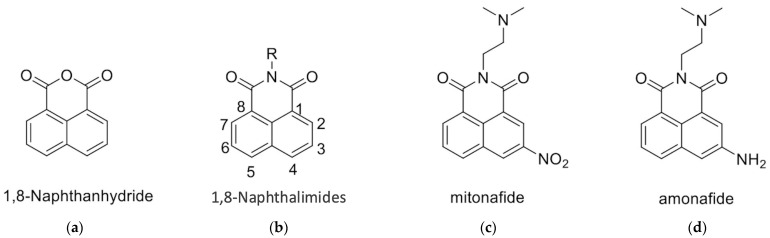
Structures of (**a**) 1,8-Naphthanhydride, (**b**) 1,8-Naphthalimide, (**c**) Mitonafide, and (**d**) Amonafide.

**Figure 2 ijms-25-11593-f002:**
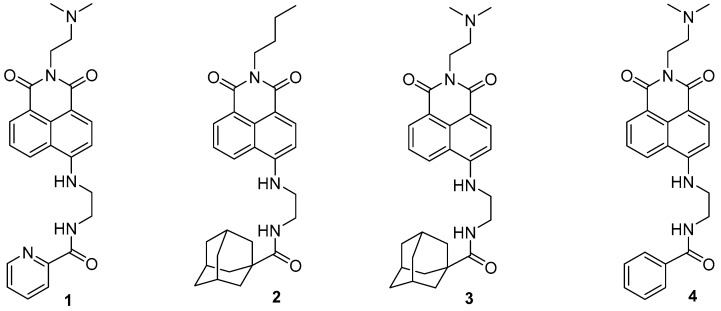
1,8-Naphthalimide derivatives **3** and **4**, used in the analyses in this study.

**Figure 3 ijms-25-11593-f003:**
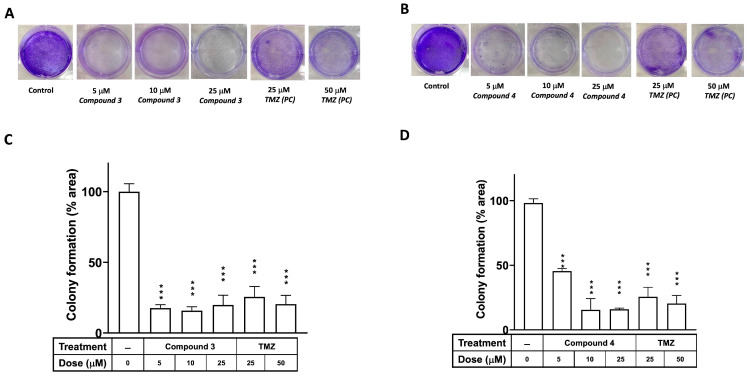
Effects of compounds **3** and **4** on clonogenic survival of U87-MG GBM cells. U87-MG cells were seeded into culture plates and treated with compound **3** (**A**) or compound **4** (**B**) at a dose of 5, 10, or 25 μM, with TMZ used as a positive control (25 or 50 μM), for 8 days. Subsequently, the cells were fixed with 1% formalin containing 1% crystal violet, and colony formation was assessed using an inverted microscope. (**C**,**D**) Colony numbers were quantified in culture plates. Data are presented as the mean ± SE of three independent experiments, and statistical analysis revealed significant differences (*** *p* < 0.001) compared with the control group.

**Figure 4 ijms-25-11593-f004:**
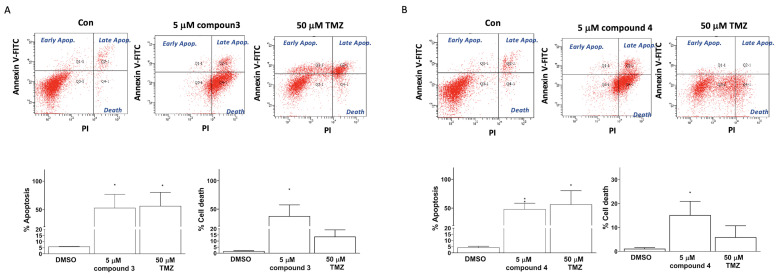
Induction of apoptosis and cell death in U87-MG GBM cells by compounds **3** and **4**. U87-MG cells were treated with either compound **3** (5 μM; **A**) or compound **4** (5 μM; **B**), with TMZ treatments at a dose of 50 μM used as a positive control. Data are presented as the mean ± SE of three independent experiments, and statistical analysis revealed significant differences (* *p* < 0.05 and ** *p* < 0.01) compared with the control group.

**Figure 5 ijms-25-11593-f005:**
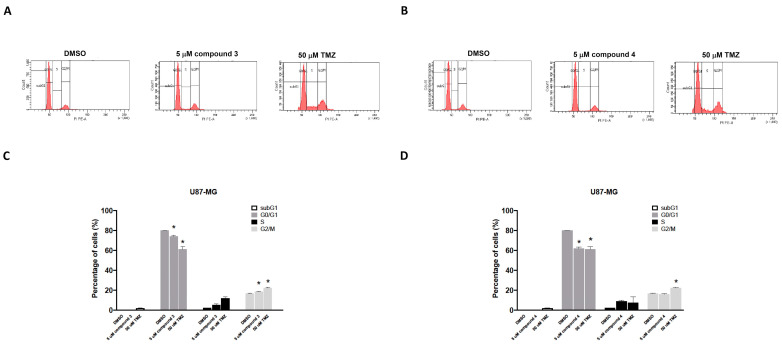
Effects of compounds **3** and **4** on the cell cycle distribution of U87-MG GBM cells in vitro. U87-MG GBM cells were treated with 5 μM of compound **3** (**A**), 5 μM of compound **4** (**B**), or 50 μM of TMZ as a positive control. After 24 h, the cells were harvested and stained with propidium iodide (PI). The percentage distribution of cells in the sub-G1, G0/G1, S, and G2/M phases was analyzed through flow cytometry (**C**,**D**). Data are presented as the mean ± SE of three independent experiments. Statistical analysis revealed significant differences (* *p* < 0.05) compared with the control group.

**Figure 6 ijms-25-11593-f006:**
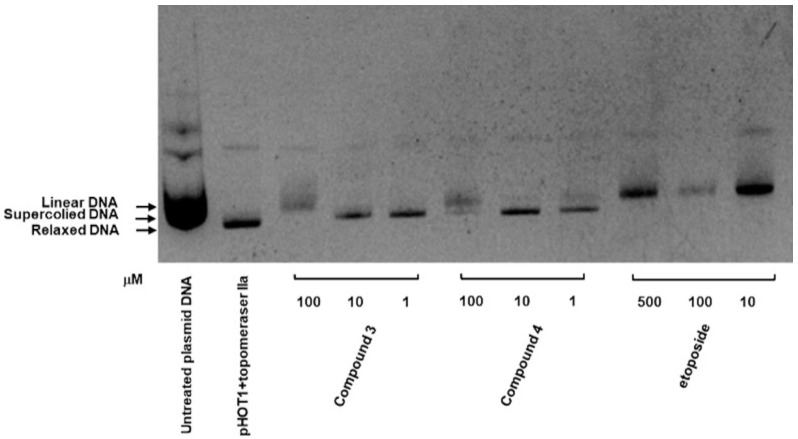
Inhibitory effect of compounds **3** and **4** on Top II. Supercoiled plasmid DNA (pHOT DNA) was incubated with Top II and various concentrations of compounds **3** and **4** (1, 10, and 100 μM) or etoposide (VP-16; 10, 100, and 500 μM). The reaction products were separated onto 1% agarose gel containing 0.5 μg/mL ethidium bromide. The experiment was independently replicated three times.

**Figure 7 ijms-25-11593-f007:**
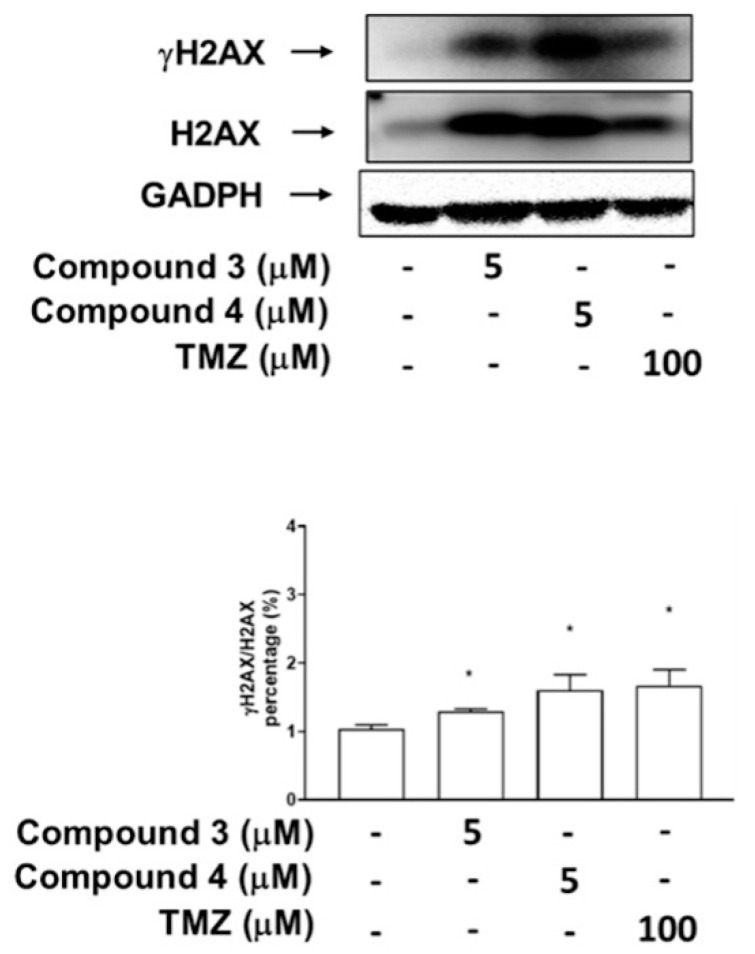
Effects of compounds **3** and **4** on phosphorylation of H2AX (DNA damage marker) in U87-MG GBM cells. The expression of H2AX and its phosphorylation status at Ser 139 (γ-H2AX) were analyzed through immunoblotting by using antibodies against the phosphorylated and total protein, with GADPH serving as the loading control. Data are presented as means ± SEs (*n* = 3). * *p* < 0.05 compared with DMSO.

**Figure 8 ijms-25-11593-f008:**
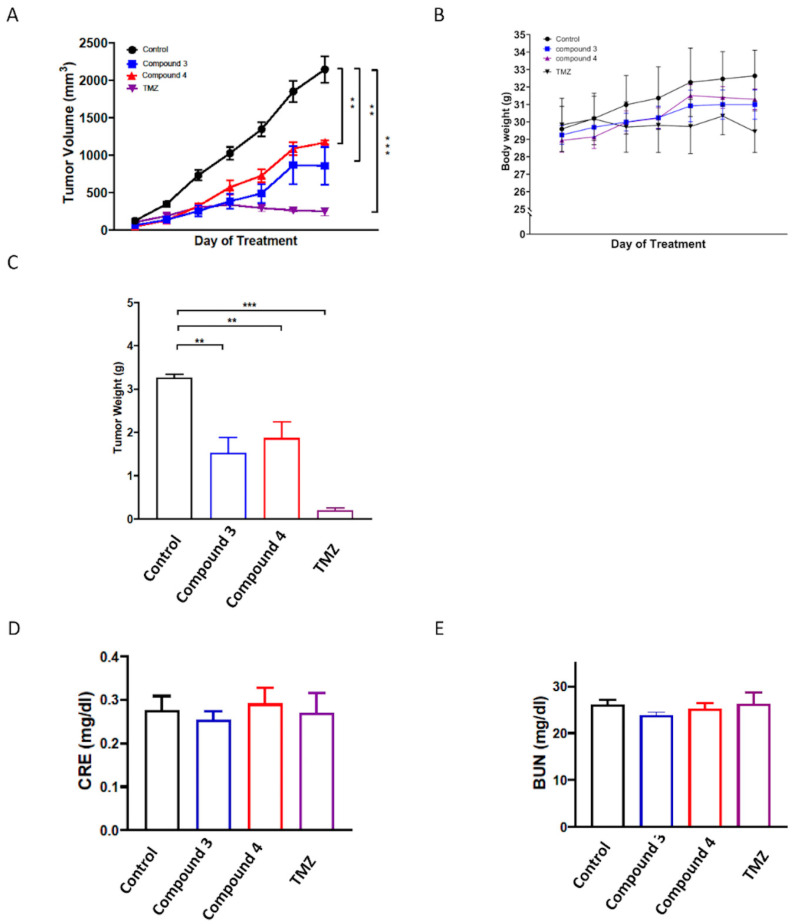
In vivo antitumor activity of compounds **3** and **4** against U87 GBM xenografts. *Nonobese diabetic* (*NOD*)/*severe combined immunodeficient* (*SCID*) mice were subcutaneously implanted with U87 GBM cells and intraperitoneally injected with the vehicle (0.1% DMSO in saline), compound **3** (5 mg/kg), compound **4** (5 mg/kg), or TMZ (12.5 mg/kg) thrice weekly. Tumor growth (**A**), body weight (**B**), and tumor weight (**C**) were recorded after the mice were euthanized. Additionally, the serum levels of (**D**) BUN and (**E**) CRE were determined using an automated clinical chemistry analyzer. Data are presented as means ± standard errors of the mean. ** *p* < 0.01 and *** *p* < 0.001 compared with the model group.

**Table 1 ijms-25-11593-t001:** Inhibitory effects of compounds **3** and **4** against human glioblastoma cells (U87-MG and DBTRG-05MG cells).

Compound	U87-MG	DBTRG-05MG
	GBM	GBM
	IC_50_ (μM) ^a^	IC_50_ (μM) ^a^
**3**	11.11 ± 1.63	5.58 ± 1.30
**4**	30.48 ± 1.86	17.32 ± 1.40
TMZ ^b^	192.20 ± 0.65	440.00 ± 1.34

Results are presented as averages ± SDs (*n* = 3). ^a^ Concentration necessary for 50% inhibition (IC_50_). ^b^ Temozolomide (TMZ) was used as a positive control.

## Data Availability

The data are contained within the article.

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
