# Peer review of "Novel 1,8-Naphthalimide Derivatives Inhibit Growth and Induce Apoptosis in Human Glioblastoma"

_ijms, 2024, doi:10.3390/ijms252111593_

Round 1
Reviewer 1 Report
Comments and Suggestions for Authors
The manuscript is very interesting however I have a few comments:
- compounds symbols are not in bold, e.g. p 5, l. 146,147,149, 151, whole paragraph 2.5 and Disucssion, as well as the title of the figures 6 and 8
- citation method needs to be corrected - In the text, reference numbers should be placed in square brackets [ ], and placed before the punctuation
- Literature format needs to be standardized, font format and journal abbreviations
#########################
• What is the main question addressed by the research?
This study presents investigations to answer whether new 1,8-naphthalimide derivatives will show suppression of GBM cells in vitro and in vivo without affecting body weight or renal toxicity. A positive response could result in a new active compound against GBM, which is a highly aggressive and invasive brain tumor associated with low patient survival rates .
survival rates..
• Do you consider the topic original or relevant to the field? Does it
address a specific gap in the field? Please also explain why this is/ is not
the case.
The described studies confirmed the suppression of GBM cells under the influence of the new compounds in vitro and in vivo, and on top of that, there was no effect of the derivatives on body weight or renal toxicity. This result is very promising and encourages further pharmacokinetic and pharmacodynamic studies.
• What does it add to the subject area compared with other published
material?
The described experiment is a contribution to the field of medicinal chemistry, allowing to develop knowledge of the structure-activity relationship. It also provides information regarding the activity testing of potential anticancer drugs. 1,8-naphthamilide derivatives may be a therapeutic alternative with potentially similar or better efficacy and fewer side effects compared to currently used chemotherapeutics.
• What specific improvements should the authors consider regarding the
methodology? What further controls should be considered?
To confirm the results, I propose detailed animal analyses, for example, MALDI-IMS (MALDI imaging mass spectrometry) analysis of tissue samples taken from experimental animals. This will make it possible to determine the composition of peptides, lipids and proteins (proteomic and metabolomic studies), as well as to assess the distribution of new compounds in the cell/tissue. Conducting these evaluations will allow us to determine the effect of the Derivatives on the cell membrane and the change in the profile of proteins/peptides and lipids in the cell after topoisomerase II inhibition. The research will reveal qualitative and quantitative differences in cells/tissues at the molecular level.
• Are the conclusions consistent with the evidence and arguments presented
and do they address the main question posed? Please also explain why this
is/is not the case.
In my opinion, the conclusions are consistent with the evidence and arguments presented. I would only add to the summary an indication of future experiments needed to complete this research work.
• Are the references appropriate?
The literature is improperly cited in the text and needs to be corrected, especially the abbreviation of journals and the type of font is not used appropriately in each item. Other than that, I believe it is selected and used appropriately.
• Any additional comments on the tables and figures.
Fonts on figure 4, 5 and 8 are too small, the whole thing is not very readable
Reviewer 2 Report
Comments and Suggestions for Authors
This article titled " Novel 1,8-Naphthamilide Derivatives Inhibit the Growth and Induce the Apoptosis in Human Glioblastoma" describes several biochemical analyses on two (new?) naphthalimide compounds (3 and 4).
My main concern starts when reading the first sentence of the abstract that is misleading and not so representative of the article.: "…we synthesized a series of novel naphthalimide derivatives with diverse substituents and investigated the effect on glioblastoma multiforme (GBM) cells.". I am still searching for the so-called "synthesis", since I did not find any, nor reference to them nor even a scheme. Moreover, when reading " a series of novel naphthalimide derivatives" and "diverse substituents", we expect to have a certain number of new compounds. The two compounds in this article, this is the minimum requirement for being called "series". Combined with the fact that we do not know where these two compounds come from, this creates questions about the article. Moreover, with so few compounds, no SAR study can be done, and the conclusion is quite short.
Another concern is that maybe the authors are biochemists, but they seem not to be synthesis chemists. First, they do not follow the rule stating that synthetized compounds should be in bold (not in bold in almost every page and SI), and they do not know how to write naphthalimide (written Naphthamilide in 13 places, including the title of the article (that should be changed) and keywords).
I also found that the results in Figure 3 are highly doubtful, since it is exactly the same picture for C and D, for the areas or their SD. Since the authors did not give the data in SI, it is impossible to check.
This article should be corrected to answer the previous comments. If the compounds are new, the synthesis should be included with the purities, since the compounds are tested in diverse biochemical tests. If not, the synthesis should be cited. I think that if the corrections are done, this article could be published, even if there are only two compounds. For these reasons, I propose that this article should be accepted after major revisions.
Comments:
Line 24-27 "…without any discernible loss of body weight or kidney toxicity noted. This study presents novel findings demonstrating that 1,8-naphthalimide derivatives exhibited significant GBM cell suppression in vitro and in vivo with no effect on body weight and no kidney toxicity.": Repetition, should be rewritten.
Line 41 "donated into the aromatic rings of 1,8-naphthamilide": I am not sure the word “donated” is the best one.
Lines 54-59 "DNA Top II … DNA topoisomerase II (Top II)": This definition should appear at the first occurrence (line 54), not line 59.
Line 70 "As shown in Figure 2, we further synthesized compounds 3 and 4 by using similar methods.": Figure 2 does not show synthesis of compounds 3 and 4 using similar methods. The synthesis is nowhere to be found. These compounds appear in Figure 2 but that is all.
Line 77 " in vitro and in vivo": In the whole text, should be in italic. The same for "ad libitum" line 353.
Figure 1 and its caption “naphthamilide": naphthalimide. The same for caption "Figure 2. 1,8-Naphthamilide derivatives 3 and 4, used in the analyses of this study." Moreover, 3 and 4 should be in bold.
Since there is no synthesis inside this paper, what is the meaning to add the anhydride into Figure 1?
Line 86 " As shown in Table 1, the IC50 values of compounds 3 and 4 were 11.11 ± 1.63 and 30.48 ± 1.86 µM, respectively.": …for U87 (not written).
Line 90 "Table 1. Inhibitory effects of compounds against…": of compounds 3 and 4 against…
Line 99 and after: 3 and 4 in bold (lines 99, 149, 151, 155, 159, 161, 162, 165, 193, 211, 212, 213, 214, 218, 220, 221, 224-227, 251-262, etc)
Line 104 " Figure 3.": It is quite strange for the SD for 25 µM to be so small for both 3 and 4. Because of that, the % for 25 µM, that seems to be higher than for 10 µM (not logical) can be lower because of the lower SD. But my main concern is that the two graphics seem to be twins (two clones). All is the same, the values (% area) and the SD(s). It makes me think of a copy/paste, when, because of the experiment’s errors, moreover, with two different cells, we expect to have differences. The authors should verify that they did not make mistakes and duplicated the same graphic.
Figure 4: Is "con" control? Since you have place to write "control", why this abbreviation?
Moreover, since you had place to write "5 µM compound 4", why writing "5 µM compoun3"?
Line 148 "topoisomerase ii": II
Line 283 "at a density of in 100 µL per well.": check this sentence.
Round 2
Reviewer 2 Report
Comments and Suggestions for Authors
The article "Novel 1,8-Naphthalimide Derivatives Inhibit the Growth and Induce the Apoptosis in Human Glioblastoma" has been modified to take into account the asked modifications. Some minor formatting problems remain but do not affect the publishability of this work.
To be sure that the new asked modifications will be taken into account, I selected "Accept after minor revisions (corrections to minor methodological errors and text editing)", because of the "text editing" part, but when done, probably there will not have the need for this article to pass through a third review.
Expected modifications:
Line 24 "cell death Only compound": Dot is missing after "death".
Lines 23, 83, 190, 196, 205, 269, 270, 401. It is strange, "in vivo" is marked as "changed", but still not in italic in my PDF.
Lines 83, 108, 154, 401 the same for in vitro.
Line 36 "through a reaction with corresponding amines, as depicted in Figure 1.": Figure 1 is not a reaction scheme using amines! But since you added the synthesis of compounds 3 and 4 in SI, without a scheme, it could be a good idea to add the reaction scheme in SI and to change this sentence to "through a reaction with corresponding amines, as depicted in supporting information".
Line 74 "As shown in Figure 2": "As shown in Figure 2 and in supporting information " (of course, only if the reaction scheme is added in SI).
Line 80 "1,8-Napththamilides": 1,8-Napththalimides
Figure 1 is marked as "modified", but "Napththamilide" is not corrected (in the figure, not in the caption that has been corrected).
Figure 2 is also marked as modified, but I cannot see what was changed…
Line 109 "As depicted in Supplementary Figure 1,": and line 231 " As shown in Figure 3 and Supplementary Figure 1". The same for Supplementary Figure 2 (line 149), 3 (lines 181 and 267) and 4 (lines 196 and 410). The file containing the cited parts, in the first submission, was replaced by the synthesis of compounds 3 and 4. I guess the old file (renamed as “ijms-3258477-peer-review-r1.docx”) will be added to SI… But why not doing only one file (figures 1-4 + synthesis and NMR) to avoid this kind of questions...?
Line 176 "compounds 3 and 4": 4 not in bold.
Figure 8 E: The 3 is a little erased in my PDF.
Line 241 " compound 3": not in bold.
Line 279 " Based on our results, we can conclude that the adamantane group is more rigid, lipophilic, stable, and efficient than the phenyl group…": Efficient, ok because 3 is more active than 4, but what is the link between your results and the stability and rigidity? I cannot see why adamantane would be more rigid and stable than a phenyl group (at least, based on your present results). If you have information about this, add it, otherwise, I propose to remove those words.
